# Temperature Reconstruction in the Southern Margin of Taklimakan Desert from *Tamarix* Cones Using GWO-SVM Model

**Zhiguang Li [1,2], Zitong Wang [1], Can Cui [1], Shuo Zhang [1] and Yuanjie Zhao [1,*]**

[1] Hebei Key Laboratory of Environmental Change and Ecological Construction, College of Geographical Sciences, Hebei Normal University, Shijiazhuang 050024, China; lizhg1978@126.com (Z.L.); 15511666716@stu.hebtu.edu.cn (Z.W.); cuican@stu.hebtu.edu.cn (C.C.); zhangs0701@126.com (S.Z.)

[2] Hebei Center for Ecological and Environmental Geology Research, Hebei GEO University, Shijiazhuang 050031, China

* Correspondence: ecoenvir@163.com

**Abstract:** The sedimentary laminae of *Tamarix* cones in arid regions are of great significance for dating and climatic reconstruction. Here, we present a multiproxy climatic record from the *Tamarix* cones in the southern margin of the Taklimakan Desert. Both the bivariate analysis and canonical correlation analysis were carried out for four groups of climate proxies in *Tamarix* cones, including organic matter content, grain size, cation content, and stable isotope content ($\delta^{13}$C, and $\delta^{18}$O). The temperature during the period from 1790 to 2010 AD has been reconstructed using the support vector machine optimized by the grey wolf optimizer, in which the climate proxies (TN, TOC, C/N, $Mg^{2+}$, $Ca^{2+}$, $\delta^{13}$C, and $\delta^{18}$O) were selected using the neighborhood rough set. The reconstructed values are in good agreement with the instrumental data. The regional temperature has distinct stages during the period from 1790 to 2010 AD, with cold conditions during 1790–1840 AD and 1896–1939 AD, and with warm conditions during 1841–1895 AD and 1940–2010 AD. The present work is beneficial to predict the future climate in the local area and encourage local governments to develop more effective measures to address the risks of climate change to environmental sustainability.

**Keywords:** *Tamarix* cones; temperature reconstruction; support vector machine; southern margin of Taklimakan Desert

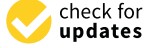



## 1. Introduction

With a typical desert–oasis ecosystem, the southern edge of the Taklimakan Desert (STD) in China plays a very important role in the implementation of the "the Belt and Road" strategy. However, the quite fragile ecology severely restricts local resource development, ecological construction, and sustainable development of economic and social ventures. Due to the close relationship between the ecological environment and climate, climate change can be used to effectively characterize the ecological environment. Indeed, paleoclimate reconstruction plays a vital role in climate prediction and then contributes to the proposal and implementation of more proactive measures to address potential environmental sustainability risks brought about by climate change, such as education, publicity, and policy adjustments to reduce carbon emissions [1].

As one special biogeomorphology, *Tamarix* cones are shrub dunes formed by the deposition of eolian sand and *Tamarix* leaves around *Tamarix* [2]. *Tamarix* cones were first introduced in 2004 for dating and environmental indication [3], then a series of studies on climate and environment have been carried out in the desert areas of Xinjiang using *Tamarix* cones as a dating tool and information carrier [4–14].

Generally, the sedimentary laminae of *Tamarix* cones have been used to study climate and environmental changes in desert areas over the past centuries or nearly a thousand years. Currently, the application of climate proxies for *Tamarix* cones is mainly in the

following two ways. The first one is to reconstruct climate sequence by correlation analysis and stepwise regression using only one single or single group of climate proxies [4–8,10–14]. Another method is to use multisingle or multigroup of climate proxies to reconstruct climate sequences through correlation analysis and stepwise regression [9].

Usually, when inferring climate based on palaeodata, it is assumed that there is a direct, linear relationship between the two. However, this relationship rarely occurs in practice. Fortunately, machine learning algorithms have significant advantages in dealing with nonlinear problems. For example, the support vector machine (SVM) has been successfully applied to runoff projection [15], the prediction of global land–ocean temperature [16], estimation for biomass [17], and imagery data analysis [18]. However, few studies have applied SVM, especially the SVM optimized by the grey wolf optimizer (GWO), to temperature reconstruction.

In this study, one GWO-SVM model for temperature reconstruction was established based on climate proxies of *Tamarix* cones in the southern edge of the Taklimakan Desert. It first utilized the neighborhood rough set (NRS) to attribute reduction for climate proxies. Then, the climate proxies selected using NRS were regarded as the input data. It is worth noting that the 29 sets of climate proxies with the corresponding instrumental temperature from 1961 to 2010 AD were selected as the training sets to establish the GWO-SVM model for temperature reconstruction. Using this model, the annual average temperature from 1790 to 2010 AD was reconstructed.

## 2. Investigated Area

The area being studied lies on the transition zone between the Taklimakan Desert and Kunlun Mountains, the upstream area of Celeriver's alluvial fan, and belongs to continental arid desert climate, with an annual average temperature of 12.13 °C and frequent wind–sand activities. According to the data gathered from Cele Meteorological Station, the annual precipitation is only 38.4 mm, 50% to 70% of which concentrates in the period from June to September; however, the annual average evaporation can reach 2500–3400 mm. The zonal vegetation is mainly made up of the xerophilous, super-xerophytic shrubs and semishrubs, as well as some alkali–saline-tolerant perennial herbs. On both sides of the riverbanks, delta areas, ancient riverbanks, and local lowlands of the rivers, there are scattered Tamarixes. The longtime collaboration between wind–sand activity and *Tamarixes* has formed *Tamarix* cones, a unique biogeographical type, composed of alternate layers of sand and *Tamarix* twigs and leaves [3]. The gravel desert is dotted with Ephedra and Sarcozygium Bunge.

## 3. Materials and Methods

### 3.1. Tamarix Cones Data

*Tamarix* cones (37.09° N, 81.08° E, 1318 m a.s.l.) are chosen in this study, which are located in the Damagou Township, Cele County, STD, Xinjiang (Figure 1). A total of 151 samples have been collected layer by layer from the top layers of *Tamarix* cones down. The climate proxies were obtained from the *Tamarix* cones samples. The age–depth models of the *Tamarix* cones have already been published [19], as shown in Figure 2. Based on laminae layers, AMS $^{14}$C dating and $^{137}$Cs dating methods, we suggested that the *Tamarix* cones formed since about 1590 AD. Due to the low resolution of the sedimentary records from 1590 to 1790 AD, here, we focus on the regional climate history of the study area during 1790 to 2010 AD.

As presented in Table 1, a total of 15 climate proxies were selected as climate indications in *Tamarix* cones, which can be divided into four groups (Table 1): (1) Organic matter content—the climate evolution can be recorded using the total nitrogen (TN), total organic carbon (TOC), and carbon–nitrogen ratio (C/N) [4,20]. (2) Grain size—the environmental information captured by grain size can not only be used to characterize the transport capabilities of wind and water but also to represent the dry and wet changes in the climate [4,21]. (3) Cation content—as important components of plants, cations significantly affect the growth and metabolism of plants by controlling the opening and closing of leaf

stomata, photosynthesis, and transpiration [22]. Climate variability is a potentially large factor influencing the dynamics of base cations ($Na^+$, $Mg^{2+}$, $K^+$, $Ca^{2+}$) in soils [23]. (4) Stable isotopic content—stable carbon isotopes and stable oxygen isotopes are mainly used to explore issues about temperature, precipitation, and monsoon intensity [24,25].

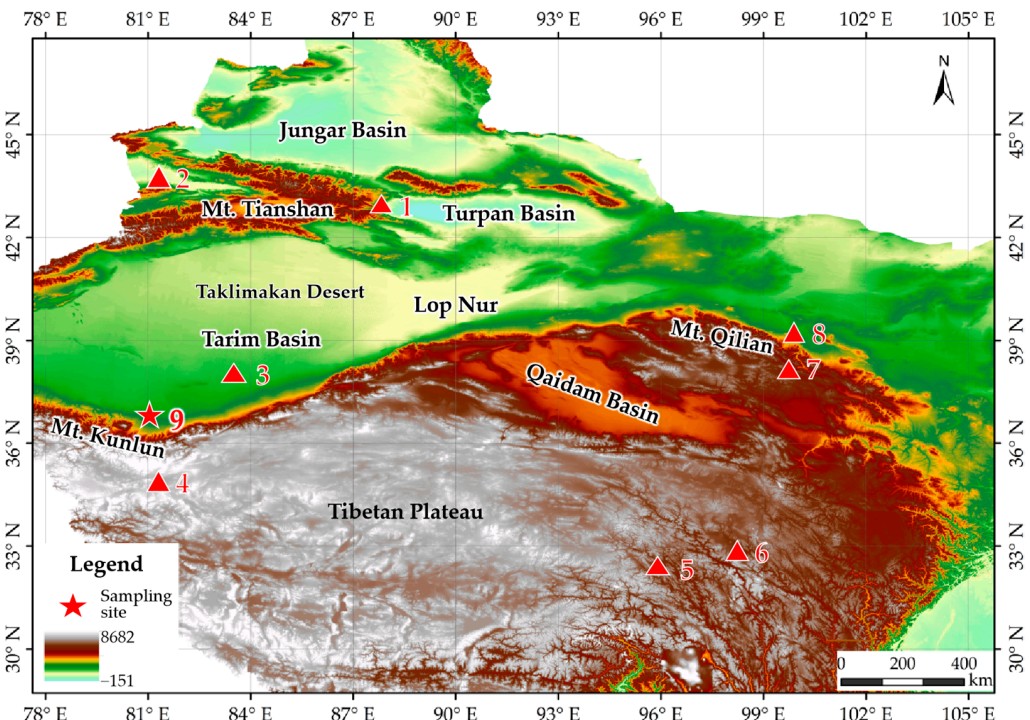

**Figure 1.** Study area, sampling sites mentioned in the text: (1) Mt. Tianshan; (2) Yili; (3) Tarim Basin; (4) Mt. Kunlun; (5) Yangtze River on the Tibetan Plateau; (6) western Sichuan Plateau; (7) Heihe River Basin; (8) Mt. Dongda region; (9) Current study area. (Sites 1, 2, 3, 8, and 9 are located in Northwest China. Sites 4, 5, 6, and 7 are located in Tibetan Plateau, China).

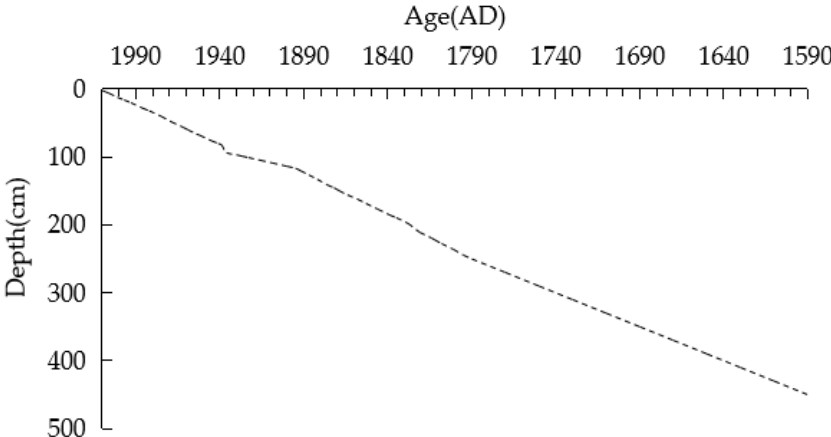

**Figure 2.** The chronological sequence of the *Tamarix* cones [19].

*3.2. Methods*

(1)    Neighborhood rough set model

As the important extensions of classical rough sets, neighborhood rough set (NRS) has significant advantages in feature selection classification modeling [26–28]. NRS can not only preserve a large amount of the key information in the original data when processing mixed data but also efficiently and accurately achieve attribute reduction. The relevant concepts and theories of NRS are as follows.

**Table 1.** Correlation matrix of temperature and climate proxies.

| Climate Proxies Group | Single Proxy |
|---|---|
| Organic matter content of *Tamarix* cones | Total nitrogen (TN)<br>Total organic carbon (TOC)<br>Carbon–nitrogen ratio (C/N) |
| Grain size of *Tamarix* cones | Average grain size ($M_z$), Median grain size ($M_d$)<br>Sorting coefficient ($S_d$), Standard deviation (S)<br>Skewness ($S_k$), Kurtosis ($K_u$) |
| Cation content of *Tamarix* cones | $Na^+$, $Mg^{2+}$, $K^+$, $Ca^{2+}$ |
| Stable isotopic content of *Tamarix* cones | $\delta^{13}C$, $\delta^{18}O$ |

For a neighborhood decision table ($NDT = <U,A,D>$), $U$ is a nonempty finite universe, which can be expressed as $U = \{x_1, x_2, \ldots x_n\}$. $A$ is the attribute set and $D$ represents the decisions. Then, the neighborhood $\delta$ for $x_n \in U$ can be described as

$$\delta_B(x_i) = \left\{ x_j \middle| x_j \in U, \Delta_B(x_i, x_j) \leq \delta \right\} \tag{1}$$

where $\Delta_B$ is the distance between $x_i$ and $x_j$ in attribute subset $B \subseteq A$. Furthermore, the upper and lower approximations of decisions $D$ to $B$ are determined as follows [29]:

$$\overline{N}_B D = \overset{n}{\underset{i=1}{U}} \overline{N}_B X_i \tag{2}$$

$$\underline{N}_B D = \overset{n}{\underset{i=1}{U}} \underline{N}_B X_i \tag{3}$$

where

$$\overline{N}_B X = \left\{ x_i \middle| \delta_B(x_i) \cap X \neq \varnothing, x_i \in U \right\} \tag{4}$$

$$\underline{N}_B X = \left\{ x_i \middle| \delta_B(x_i) \subseteq X, x_i \in U \right\} \tag{5}$$

The decision system boundary can also be obtained as

$$BN(D) = \overline{N}_B D - \underline{N}_B D \tag{6}$$

It should be noted that the lower approximations are the same as the positive region of the decision ($P_B(D)$):

$$P_B(D) = \underline{N}_B D \tag{7}$$

From the $P_B(D)$, the dependence of $D$ on $B$ can be calculated as follows:

$$\gamma_B(D) = \frac{|P_B(D)|}{U} \tag{8}$$

where $\gamma_B(D)$ is the dependence of $D$ on $B$. Moreover, the relationship between $\gamma_B(D)$ and the importance of $D$ to $B$ can be expressed as

$$S_{ig}(a, B, D) = \gamma_{B \cup \{a\}}(D) - \gamma_B(D) \tag{9}$$

where $a \subseteq B - A$. $S_{ig}(a,B,D)$ is the importance of $a$ to $B$.

(2) Support vector machine optimized by grey wolf optimizer

The support vector machine (SVM) is a machine learning method that can be used to address the binary classification problem. For a given test set, it can be expressed as $\{(x_i, y_i), i = 1, 2, \ldots, l, y_i = +1 \text{ or } -1\}$, where l refers to the number of cases. The key to solving the

classification problem is to find the relevant hyperplane [30], as shown in Figure 3. The optimal hyperplane can be determined as follows [31]:

$$\omega^T x + b = 0 \tag{10}$$

where $w$ is a normal vector and $b$ is the bias. If the inequality of Equations (11a) and (11b) are satisfied, the sample set is considered linearly separable:

$$\omega^T x_i + b \geq 1, y_i = +1 \tag{11a}$$

$$\omega^T x_i + b \leq 1, y_i = -1 \tag{11b}$$

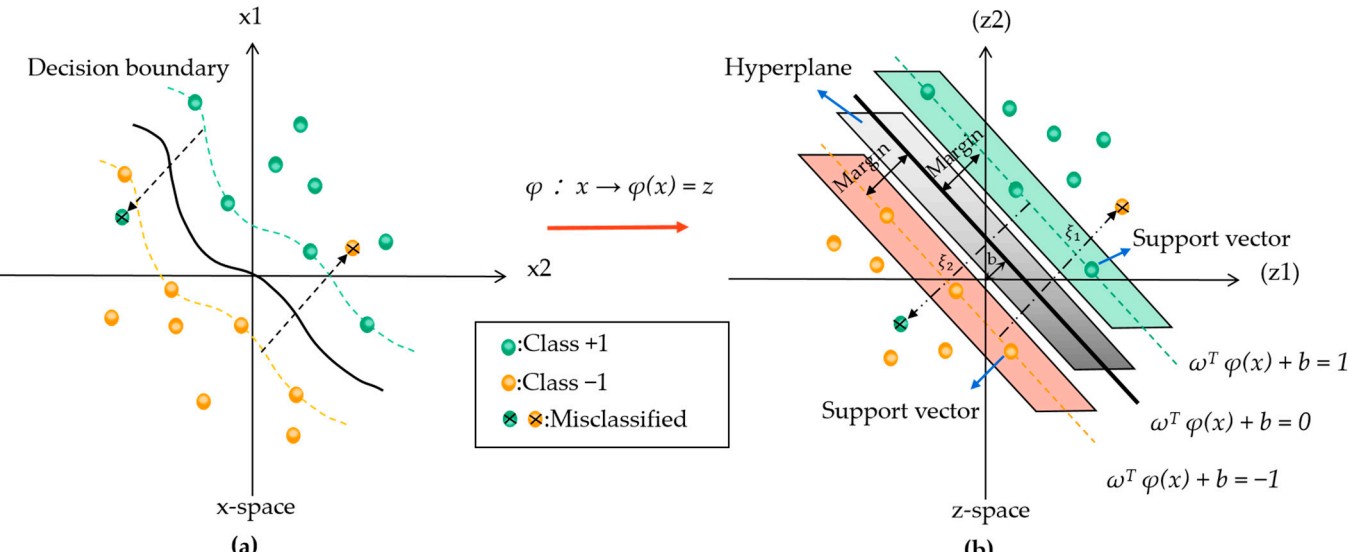

**Figure 3.** Graphical representation of support vector machine: (**a**) Complex nonlinear classification problem representation in low-dimensional space; (**b**) Linear classification problem representation in high-dimensional space [30].

The distances between the hyperplane and the support vectors are expressed as

$$d = \frac{2}{\|\omega\|} \tag{12}$$

When the samples are linearly separable, the optimal hyperplane problem can be described as the following constrained optimization problem:

$$\min_{\omega, b} \ \frac{\|\omega\|^2}{2} \tag{13}$$

$$s.t. y_i(\omega^T x_i + b) \geq 1, i = 1, 2, 3, \dots, l$$

In the case that the samples are linearly nonseparable, the corresponding optimization problem can be described as

$$\min_{\omega, b} \ \frac{\|\omega\|^2}{2} + C \sum_{i=1}^{l} \xi_i \tag{14}$$

$$s.t. y_i(\omega^T x_i + b) \geq 1 - \xi_i, \xi_i \geq 0, i = 1, 2, 3, \dots, l$$

where $C$ is the penalty factor and $\xi_i$ is the slack variable.

According to the Lagrange multiplier method, the Lagrange objective function is constructed as [32]

$$L_p = \frac{\|\omega\|^2}{2} + C \sum_{i=1}^{l} \xi_i - \sum_{i=1}^{l} \alpha_i \left[ y_i(\omega^T x_i + b) - 1 + \xi_i \right] - \sum_{i=1}^{l} \beta_i \xi_i \qquad (15)$$

where $\alpha_i$ and $\beta_i$ are Lagrange coefficients.

Finally, the optimal decision function is obtained:

$$f(x) = sign \left[ \sum_{i=1}^{l} y_i \alpha_i K(x \cdot x_i) + b \right] \qquad (16)$$

where $K(x \cdot x_i)$ is kernel function. In this paper the radial basis function (RBF) was selected as the kernel function considering its lower complexity.

However, it is difficult to obtain the penalty factor $C$ and kernel parameter g. To improve the performance of the model, a grey wolf optimizer (GWO) was used to optimize the parameters of SVM. The GWO model can be mathematically described as social hierarchy and hunting [30]. In the social hierarchy of a grey wolf population (Figure 4), the leading wolf ($\alpha$) at the top of the pyramid is the optimal solution. From top to bottom, $\beta$, $\delta$, and $\omega$ are the second, third, and alternative solutions, respectively. The solving process of GWO mainly includes encircling prey and hunting. The update of the grey wolf position in GWO (Figure 5) can be expressed as

$$\vec{X}(t+1) = \frac{1}{3}\vec{X}_1 + \frac{1}{3}\vec{X}_2 + \frac{1}{3}\vec{X}_3 \qquad (17)$$

$$\vec{X}_1 = \vec{X}_\alpha(t) - \vec{A}_1 \cdot \vec{D}_\alpha, \vec{X}_2 = \vec{X}_\beta(t) - \vec{A}_2 \cdot \vec{D}_\beta, \vec{X}_3 = \vec{X}_\delta(t) - \vec{A}_3 \cdot \vec{D}_\delta \qquad (18)$$

$$\vec{D}_\alpha = \left| \vec{C}_1 \cdot \vec{X}_\alpha(t) - \vec{X} \right|, \vec{D}_\beta = \left| \vec{C}_2 \cdot \vec{X}_\beta(t) - \vec{X} \right|, \vec{D}_\delta = \left| \vec{C}_3 \cdot \vec{X}_\delta(t) - \vec{X} \right| \qquad (19)$$

where $\vec{X}_1$, $\vec{X}_2$, and $\vec{X}_3$ represent the positions of $\alpha$, $\beta$, and $\delta$. $\vec{X}(t+1)$ is the position of $\omega$ at the time of $(t + 1)$. $\vec{D}_\alpha$, $\vec{D}_\beta$, and $\vec{D}_\delta$ are the distances between $\omega$ and $\alpha$, $\beta$ and $\delta$, respectively.

(3)　Statistical analyses

In this paper, the correlations between climate proxies and temperature were determined using Pearson correlation analyses (PCA) and Canonical correlation analyses (CCA) from IBM SPSS Statistics 2.4, respectively.

The results of the PCA are expressed in the Pearson correlation coefficient ($r$), which is a statistical indicator reflecting the degree of linear correlation between two variables:

$$r = \frac{\sum (X - \overline{X})(Y - \overline{Y})}{\sqrt{\sum (X - \overline{X})^2 \sum (Y - \overline{Y})^2}} \qquad (20)$$

where $X$ (or $Y$) is variable and $X$ (or $Y$) is the mean value of $X$ (or $Y$). As a dimensionless statistical indicator, the range of correlation coefficient values is $-1 \leq r \leq 1$. Moreover, when $r$ is greater or less than zero, it indicates a positive or negative correlation between two variables. Specially, the Pearson correlation coefficient $r = 0$ suggests no association.

CCA is a multivariate statistical analysis method that uses the correlation between comprehensive variable pairs to reflect the overall correlation between two groups of

variables. For two given vectors $X = (x_1, \ldots, x_n)$ and $Y = (y_1, \ldots, y_n)$, linear combinations of $X_i$ and $Y_i$ will be identified first.

$$U = c' \sum_{XX}^{-1/2} X = a'X, V == d' \sum_{YY}^{-1/2} Y = b'Y \tag{21}$$

where $U$ and $V$ are the first linear combinations, namely the first pair of canonical variables. The obtained vectors $a$ and $b$ maximize the correlation between random variables $a'X$ and $b'Y$. Then, using the correlation between these two linear combinations ($U$ and $V$) describes the overall between $X$ and $Y$.

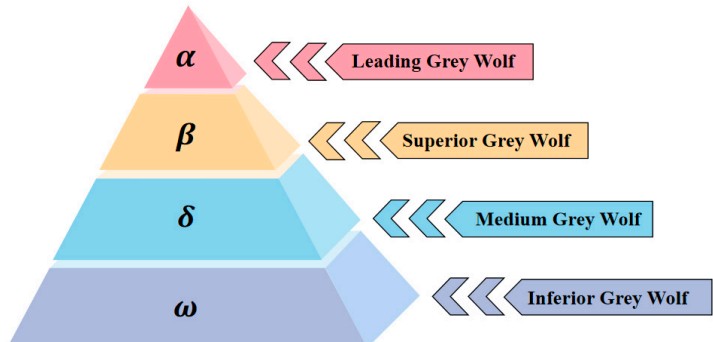

**Figure 4.** The social hierarchy of grey wolves [30].

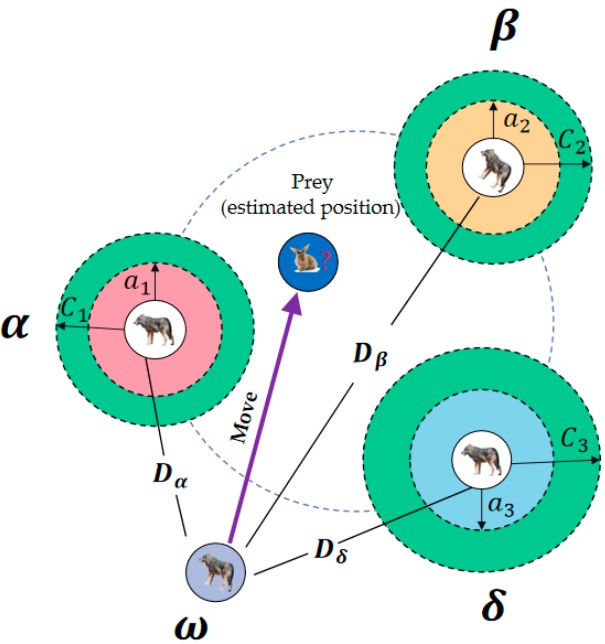

**Figure 5.** Schematic diagram of the update of the grey wolf position in GWO [30].

## 4. Reconstruction of Annual Mean Temperature

### 4.1. The Correlation between the Climate Proxies and Their Relationships with Temperature

The 29 sets of climate proxies from 1961 to 2010 AD used in the correlation analysis refer to the published data by Zhang et al. [9]. The corresponding instrumental temperature for these climate proxies were provided by the Xinjiang Cele Meteorological Station (37.02° N, 80.80° E, 1336.5 m a.s.l.), which is approximately 24 km away from the sampling sites. The Pearson correlation coefficient for the above climate proxies and instrumental data are detailed in Table 2.

**Table 2.** Correlation matrix of temperature and climate proxies. TN is total nitrogen; TOC is total organic carbon; C/N is carbon–nitrogen ratio; $M_z$ is average grain size; $M_d$ is median grain size; $S_d$ is sorting coefficient; S is standard deviation; $S_k$ is skewness; and $K_u$ is kurtosis.

| | TN | TOC | C/N | $M_z$ | $M_d$ | $S_d$ | S | $S_k$ | $K_u$ | $Na^+$ | $Mg^{2+}$ | $K^+$ | $Ca^{2+}$ | $\delta^{13}C$ | $\delta^{18}O$ | Temp. |
|---|---|---|---|---|---|---|---|---|---|---|---|---|---|---|---|---|
| TN | 1 | - | - | - | - | - | - | - | - | - | - | - | - | - | - | - |
| TOC | 0.11 | 1 | - | - | - | - | - | - | - | - | - | - | - | - | - | - |
| C/N | −0.89** | 0.21 | 1 | - | - | - | - | - | - | - | - | - | - | - | - | - |
| $M_z$ | −0.19 | 0.05 | 0.21 | 1 | - | - | - | - | - | - | - | - | - | - | - | - |
| $M_d$ | −0.17 | 0.05 | 0.20 | 0.99** | 1 | - | - | - | - | - | - | - | - | - | - | - |
| $S_d$ | −0.59** | −0.24 | 0.47** | 0.18 | 0.16 | 1 | - | - | - | - | - | - | - | - | - | - |
| S | 0.53** | 0.15 | −0.45* | −0.73** | −0.71** | −0.79** | 1 | - | - | - | - | - | - | - | - | - |
| $S_k$ | 0.05 | −0.07 | −0.08 | −0.91** | −0.90** | 0.09 | 0.54** | 1 | - | - | - | - | - | - | - | - |
| $K_u$ | −0.47** | −0.14 | 0.41* | 0.61** | 0.59** | 0.67** | −0.89** | −0.54** | 1 | - | - | - | - | - | - | - |
| $Na^+$ | −0.09 | 0.37 | 0.33 | −0.10 | −0.10 | −0.01 | 0.06 | 0.06 | 0.06 | 1 | - | - | - | - | - | - |
| $Mg^{2+}$ | 0.30 | 0.01 | −0.19 | −0.06 | −0.05 | −0.32 | 0.22 | −0.1 | −0.02 | 0.57** | 1 | - | - | - | - | - |
| $K^+$ | 0.35 | 0.31 | −0.15 | −0.16 | −0.15 | −0.38* | 0.36 | 0.03 | −0.24 | 0.74** | 0.84** | 1 | - | - | - | - |
| $Ca^{2+}$ | −0.18 | −0.16 | 0.14 | 0.29 | 0.30 | 0.32 | −0.42* | −0.24 | 0.46* | 0.07 | 0.43* | 0.15 | 1 | - | - | - |
| $\delta^{13}C$ | −0.30 | 0.32 | 0.38* | 0.08 | 0.09 | 0.23 | −0.13 | 0.13 | −0.11 | −0.10 | −0.34 | −0.23 | −0.04 | 1 | - | - |
| $\delta^{18}O$ | 0.03 | 0.42* | 0.13 | −0.15 | −0.15 | −0.03 | 0.08 | 0.08 | 0.07 | 0.39* | 0.23 | 0.21 | 0.06 | −0.16 | 1 | - |
| Temp. | −0.26 | −0.12 | 0.15 | 0.24 | 0.24 | 0.36 | −0.42* | −0.16 | 0.38* | −0.34 | −0.51** | −0.63** | 0.06 | 0.12 | −0.11 | 1 |

\* Significant at $p < 0.05$, \*\* Significant at $p < 0.01$.

For the climate proxies in the organic matter content group, TN is significantly negatively correlated with C/N, $S_d$, and $K_u$, respectively, while positively correlated with S. Furthermore, the highly positive correlations of TOC with $\delta^{18}O$ were observed. In addition, there is a strongly positive correlation between C/N and other proxies, including $S_d$, $K_u$, and $\delta^{13}C$, and a negative correlation between C/N and S. The above facts indicate that not only do most climate proxies in the organic matter content group correlate strongly with each other but also correlate well with the climate proxies in the grain size group.

For the climate proxies in the grain size group, as detailed in Table 2, $M_z$ significantly positively correlates with $M_d$ and $K_u$ but negatively correlates with S and $S_k$. Similarly, the correlation between $M_d$ and $K_u$ is obviously positive, while $M_d$ negatively correlates with S and $S_k$. In addition, $S_d$ significantly positively correlates with $M_d$ and $K_u$, while negatively correlates with S and $K^+$. There is a strongly positive correlation between S and $S_k$, and the correlation for S with $K_u$ and $Ca^{2+}$ is clearly negative. Furthermore, the highly negative correlation between $S_k$ and $K_u$ and the positive correlation between $K_u$ and $Ca^{2+}$ were found. These relationships suggest that the climate proxies in the grain size group correlate significantly but also correlate well with the climate proxies in the cation content group.

For the climate proxies in the cation content group, $Na^+$ apparently positively correlates with $Mg^{2+}$, $K^+$, and $\delta^{18}O$, respectively. Further, the obviously positive correlation can also be observed for $Mg^{2+}$ with $K^+$ and $\delta^{18}O$. This means that there is an apparent correlation between climate proxies (except for $Ca^{2+}$) in the cation content group, which also significantly correlates with climate proxies in the stable isotopic content group. On the other hand, $\delta^{13}C$ and $\delta^{18}O$ are basically unrelated.

It is worth noting that the annual mean temperature significantly negatively correlates with S, $Mg^{2+}$, and $K^+$, respectively, but positively correlates with $K_u$. The above analysis suggests that S, $K_u$, $Mg^{2+}$, and $K^+$ (or the other climate proxies that obviously correlate

with these proxies) can be preliminarily selected as the optimized proxies to reconstruct the annual mean temperature.

### 4.2. The Correlation between the Climate Proxies Groups and Their Relationships with Temperature

Canonical correlation analyses between the climate proxies groups and their relationships with annual mean temperature are detailed in Tables 3 and 4. Due to both the fewer climate proxies in the stable isotopic content group and the weaker correlation between them and the climate proxies in other groups, only the correlations between the groups of organic matter content, grain size, and cation content were analyzed.

**Table 3.** The correlation between groups of climate proxies.

| | Correlation Coefficient | Significance |
|---|---|---|
| Organic matter content vs. Grain size | 0.661 | 0.521 |
| Organic matter content vs. Cation content | 0.721 | 0.002 ** |
| Grain size vs. Cation content | 0.774 | 0.031 * |

* Significant at $p < 0.05$, ** Significant at $p < 0.01$.

**Table 4.** Correlation between climate proxies group and annual mean temperature.

| | Correlation Coefficient | Significance |
|---|---|---|
| Organic matter content | 0.313 | 0.454 |
| Grain size | 0.748 | 0.003 ** |
| Cation content | 0.674 | 0.004 ** |
| Stable isotopic content | 0.310 | 0.001 ** |

** Significant at $p < 0.01$.

In Table 3, the results suggest that there is no significant correlation between the organic matter content group and the grain size group, which is inconsistent with the correlation between the climate proxies in these two groups (Table 2). Additionally, the organic matter content group highly correlates with the cation content group, which is not identical to the correlation between the climate proxies in these two groups. Interestingly, the significant correlation between the grain size group and the cation content group agreed with the correlation between the climate proxies in these two groups.

As presented in Table 4, there is no significant correlation between the organic matter content and the annual mean temperature, while both the grain size and cation content significantly correlate with the annual mean temperature. This fact is consistent with the optimization results by the correlation between climate proxies. Nonetheless, the significant correlation between the stable isotopic content and the annual mean temperature does not correspond to the optimization results based on the correlation between climate proxies.

Thus, it can be inferred that there is a complex correlation between climate proxies and temperature. Inferring climate from palaeodata frequently assumes a direct, linear relationship between the two, which is seldom met in practice [33–36]. Therefore, reasonably selecting modeling parameters from the above climate proxies is the key to reconstructing temperature.

### 4.3. Proxies Selection

Given the situation of data collection for *Tamarix* cones in STD, 15 climate proxies were used as the attribute values of NRS, and annual mean temperature is considered as the decision. Based on NRS theory, the dependence of decision on attribute sets was determined, and the importance of each attribute in its own set was analyzed. Finally, attributes with an importance greater than zero were selected to obtain a reduction set, which contains factors that are sensitive to temperature changes.

The importance of climate proxies to the annual mean temperature is shown in Figure 6. Obviously, the importance of TN, TOC, C/N, $Mg^{2+}$, $Ca^+$, $\delta^{13}C$, and $\delta^{18}O$ is greater than

zero. Thus, these climate proxies were selected to reconstruct the annual mean temperature. It is well known that the intensity of photosynthesis and plant growth affects TN and TOC. Further, both photosynthesis and plant growth are closely related to temperature. As a result, temperature affects TN and TOC. As an important component of the cell wall and chlorophyll, $Ca^+$ and $Mg^{2+}$ are intrinsically related to the growth of *Tamarix*. Therefore, $Ca^+$ and $Mg^{2+}$ are influenced by temperature. Although $Na^+$ and $K^+$ are not selected as the optimal proxies, both of them highly correlate with $Mg^{2+}$. This indicates that $Na^+$ and $K^+$ can be quantitatively described by $Mg^{2+}$. The fractionations of $\delta^{13}C$ and $\delta^{18}O$ are related to the activity of enzymes in plants. Thus, $\delta^{13}C$ and $\delta^{18}O$ in *Tamarix* cones are significantly influenced by temperature. The climate proxies of grain size are not considered as optimal proxies. The reason for this is that these proxies are mainly controlled by wind dynamic conditions but are less affected by temperature.

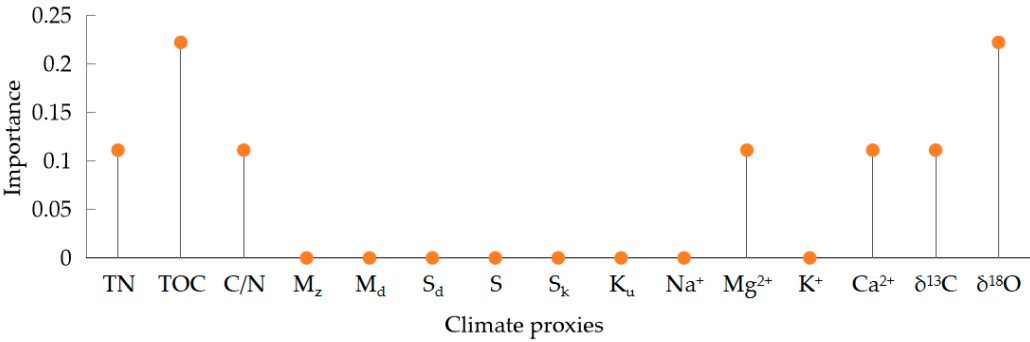

**Figure 6.** The importance of climate proxies to the annual average temperature. TN is total nitrogen; TOC is total organic carbon; C/N is carbon–nitrogen ratio; $M_z$ is average grain size; $M_d$ is median grain size; $S_d$ is sorting coefficient; S is standard deviation; $S_k$ is skewness; and $K_u$ is kurtosis.

In summary, using the NRS theory, the cross redundancy between climate proxies and between each of them and the annual mean temperature were well addressed, and the optimization of climate proxies can be reasonably explained.

*4.4. Establishment of the Model*

In the present work, 29 sets of instrumental data from 1961 to 2010 AD are used as training samples. The input data for the model established in this paper are TN, TOC, C/N, $Mg^{2+}$, $Ca^+$, $\delta^{13}C$, and $\delta^{18}O$ obtained using the attribute reduction in NRS, and the annual mean temperature is taken as output data. According to the grey wolf optimizer (GWO) algorithm, the penalty factor C and kernel parameter g were determined to be 1.21 and 0.05, respectively. After training the temperature reconstruction model based on the training samples, the reconstructed values of annual mean temperature covering 1961–2010 AD were obtained.

The comparison between the reconstructed and instrumental temperature is shown in Figure 7. Obviously, the reconstructed values are in good agreement with the instrumental data. For annual mean temperature, although there are some deviations between the reconstructed and instrumental values, the adjusted $R^2$ and correlation are 0.67 and 0.95, respectively. In other words, the reconstruction strongly correlates with the records. In addition, the trends of reconstructed and instrumental temperature are basically the same. It can be found that the annual mean temperature shows an overall ascending trend during the period of 1961–2010 AD. On the other hand, the stepwise regression method has also been attempted to establish models, wherein the adjusted $R^2$ and correlation are 0.39 and 0.63, which are significantly lower than those of GWO-SVM. This means that the temperature reconstruction of the GWO-SVM model is more reasonable than that of the stepwise regression model (Figure 7).

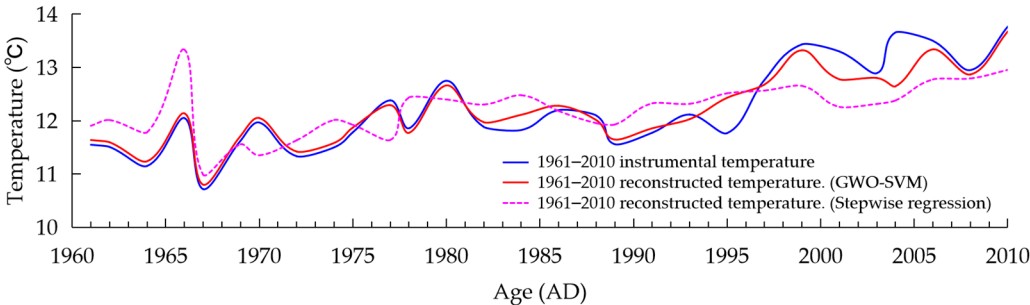

**Figure 7.** Comparison between the reconstruction temperature and the instrumental temperature.

## 5. Results and Discussion

*5.1. Overall Change and Stage Division of Regional Annual Mean Temperature*

Based on the GWO-SVM model, the annual mean temperature covering 1790–2010 AD was reconstructed in STD (Figure 8a). The mean, lowest, and highest temperatures from 1790 to 2010 AD in STD are 11.7 °C, 10.3 °C (1795 AD), and 13.7 °C (2010 AD), respectively. In general, the warmer and colder years were analyzed by whether the annual mean temperature was greater than ±1 standard deviation of the mean temperature of the full period [37,38]. The standard deviation of the reconstructed temperature is 0.56 °C. The colder years (<(mean − 1σ), namely 11.11 °C) in the reconstruction mainly occurred in the late 18th century and the first 20 years of the 20th century. Moreover, the warmer years (>(mean + 1σ), namely 12.33 °C) mainly occurred during the 1940s, 1950s, and the period from 1995 to 2010 AD. The evident cold periods are centered on the beginning of the 20th century, and the warmer period appears during the period of 1995 to 2010 AD. Interestingly, as can be seen from the instrumental temperature, the warmer climate continues after 2010 AD [39].

Figure 8b presents the reconstructed temperature smoothed with 10-year moving average. Since 1790 AD, the annual mean temperature in STD has shown an upward trend, which is similar to the change of global climate and climate in Xinjiang since the Little Ice Age [40]. As shown in Figure 8b, from the end of the 18th century to the 1880s, the annual mean temperature was in a fluctuating and rising stage. However, the annual mean temperature decreased significantly around the 1890s. After that, the annual mean temperature was in a small and frequent fluctuation state until the end of the 1930s. Then, the annual mean temperature in STD fluctuated and increased again.

The reconstructed annual mean temperature cumulative anomaly curve is presented in Figure 8c. If the annual mean temperature is higher or lower than the mean temperature of the entire period, the curve correspondingly shows ascending or descending trends, respectively. Ignoring the fluctuations with small magnitude and timing, the period on the descending and ascending segment of the cumulative anomaly curve represents a cold and warm period [11,14], respectively. Thus, as shown in Figure 8c, there are obviously two cold periods (1790–1840 AD and 1896–1939 AD) and two warm periods (1841–1895 AD and 1940–2010 AD) in STD covering 1790–2010 AD. During the cold periods with a magnitude of 10.3–12.5 °C, the mean temperature (11.4 °C and 11.3 °C for 1790–1840 AD and 1896–1939 AD, respectively) is lower than that of the whole period (11.7 °C). For the warm periods, the highest and lowest temperatures are 13.7 °C and 10.8 °C, as presented in Figure 8a. As expected, the mean temperature of these two warm periods (12.2 °C and 11.8 °C for 1841–1895 AD and 1940–2010 AD, respectively) is higher than that of the whole period (11.7 °C).

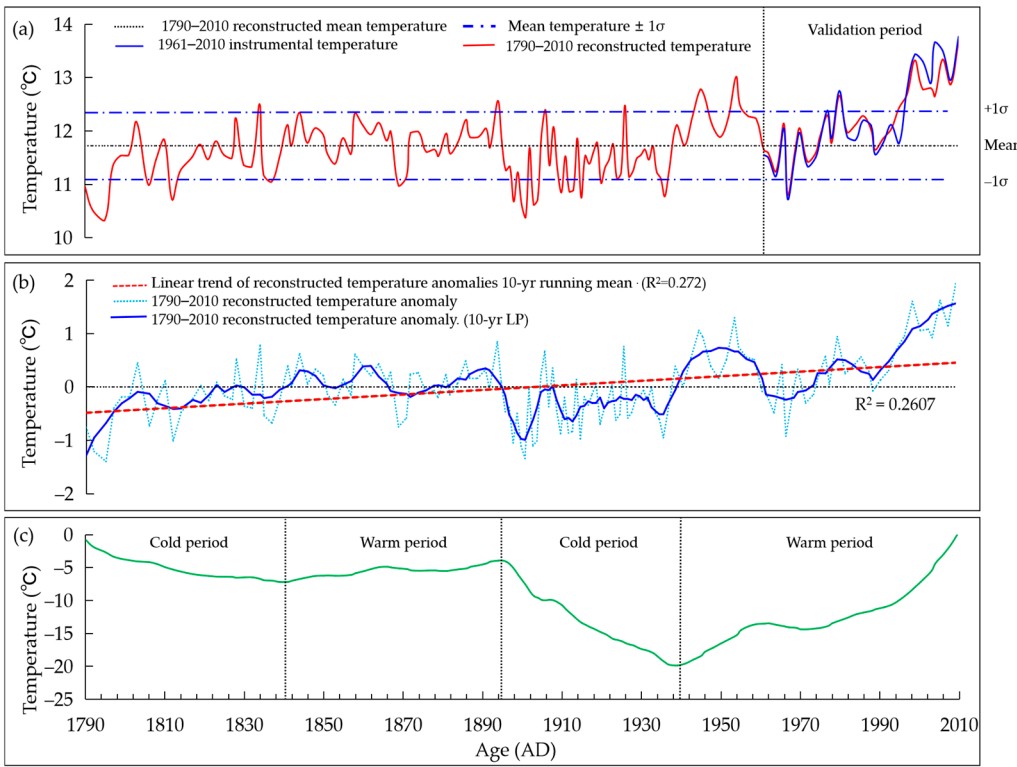

**Figure 8.** Annual mean temperature and anomaly analysis: (**a**) Reconstructed annual mean temperature series. (**b**) Reconstructed annual mean temperature anomalies. (**c**) Reconstructed annual mean temperature cumulative anomaly.

*5.2. Comparison with Other Temperature Reconstructions around the STD*

To verify the reliability of the temperature series reconstructed in this paper, Figure 9 shows the temperature reconstruction from *Tamarix* cones and other temperature reconstructions around the region from tree rings. The temperature reconstruction in this study coincides with the reconstructed annual mean temperature of the Heihe River Basin (Figure 9a) [41], reconstructed temperature for Mountain Dongda region (Figure 9b) [37], reconstructed June-July temperature since 1383 AD for western Sichuan Plateau (Figure 9c) [42], reconstructed summer temperature for the source region of the Yangtze River on the Tibetan Plateau (Figure 9d) [43], and reconstructed mean temperature for the region of Xinjiang (Figure 9e) [44]. The above comparison indicates that these reconstructed temperatures share the same trend, i.e., the warming from the 19th to the 20th century. Further, three temperature records (including the temperature reconstruction in the present work) from the late 20th century to the early 21st century show that the rate of warming is quite high from 1990 to 2010 AD, which can be also confirmed by the reconstructed summer temperature for eastern Tibetan Plateau using tree rings [45]. It is worth noting that both the same cold period from 1910 to 1927 AD and the same warm period from 1939 to 1959 AD occur in these five reconstructions (including the temperature reconstruction in the present work). Four of these reconstructions (including the temperature reconstruction in the present work) share the same cold period from 1790 to 1815 AD and the same warm period from 1886 to 1895 AD. In addition, these four reconstructed temperature curves also have multiple peaks and troughs that occur in similar years.

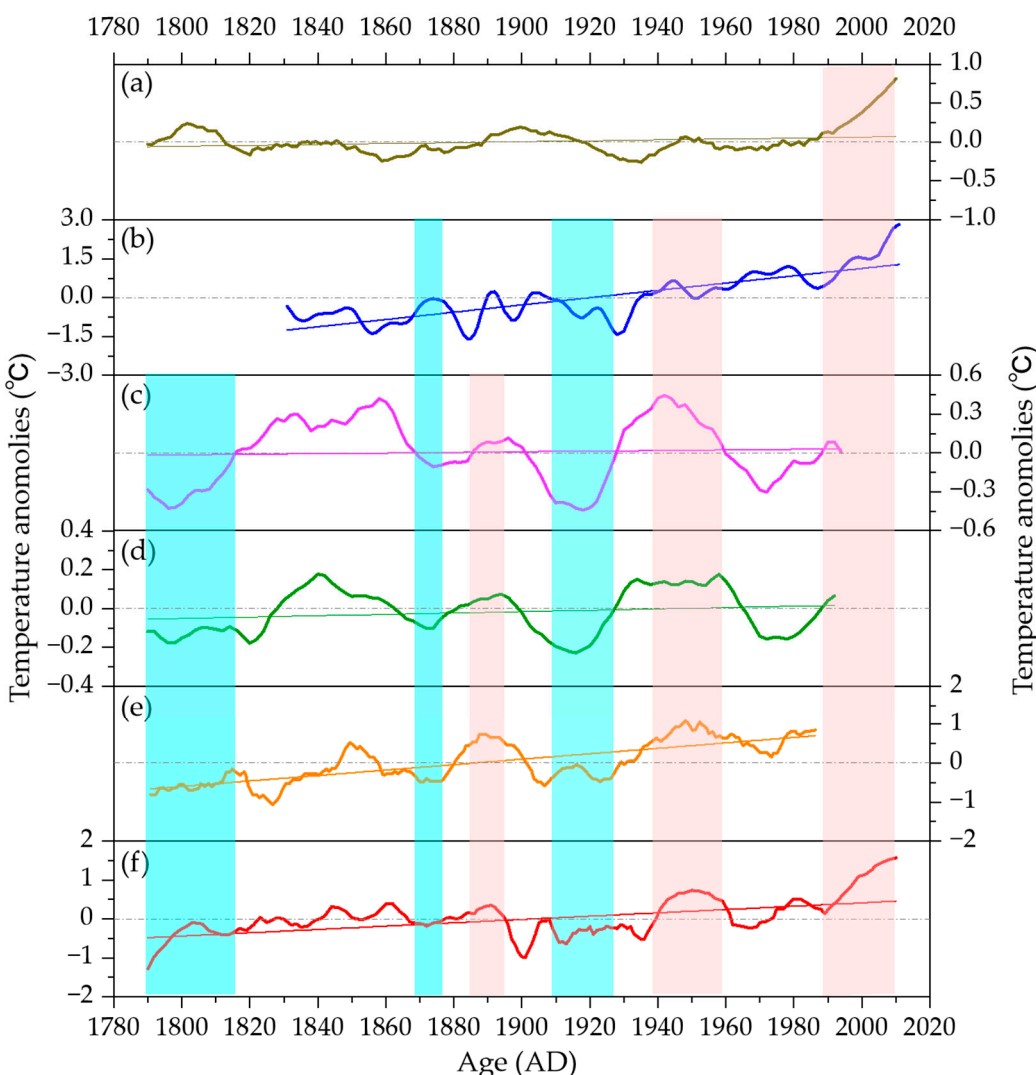

**Figure 9.** Comparison of the current study reconstructed temperature: (**a**) Wang et al. (2016) tree-ring-based reconstruction of temperature variability for the Heihe River Basin [41]; (**b**) Liu et al. (2016) tree-ring-based temperature Reconstruction for the Mt. Dongda region [37]; (**c**) Zhang et al. (2022) reconstruction of June–July temperature for the western Sichuan Plateau using tree-ring width [42]; (**d**) Liang et al. (2008) tree-ring-based summer temperature reconstruction for the source region of the Yangtze River on the Tibetan Plateau [43]; (**e**) Wang et al. (1996) changes in temperature for the Yili area from tree rings [44]; (**f**) Reconstruction of annual mean temperature in the current study for the southern margin of the Taklimakan Desert from *Tamarix* cones. The reconstructed series were all smoothed with a 10-year low-pass filter. The shaded areas indicate periods of similar trends among the reconstructions. The straight lines are trend lines for temperature reconstructions.

Moreover, several typical climate events have also been well reflected in our reconstructed temperature series. For example, according to the "Table of Natural and Manmade Disasters in the Past Dynasties of China", the northern Xinjiang and the eastern Tianshan Mountains experienced a high-temperature climate in 1952 AD [44]. Interestingly, the time of this event almost coincides with the peak of 1950 AD in the temperature reconstruction curve obtained in this study. In other words, the peak of 1950 AD may have captured the aforementioned high-temperature event. A cold event recorded through the Guriya ice core (presumably due to the eruption of Tambora volcano) [46] coincides with the trough of 1820 AD in the reconstructed temperature. Based on the Guliya ice core, the Tibetan Plateau experienced a warm period in the 1950s, followed by a cold period with the lowest temperature in 1969 AD, and then began to warm up in the 1980s [47]. This trend is highly

consistent with the changes of the period (1940–1990 AD) in the reconstructed temperature series of the current work. The arid region of Central and East Asia experienced low temperatures in the early 19th century, and then began to warm up in 1920s, which is also reflected in the eastern monsoon region, the arid region of Central Asia, and the arid region of the Mongolian Plateau. Compared to these regions, this warming trend in arid areas of the Tarim (sampling sites for this study are located at the edge of this region) appeared 20 years later (namely 1940 AD) [48], which is in good agreement with the temperature series reconstructed in this study. In summary, our temperature reconstruction is consistent with other temperature reconstructions around STD, and the reliability of the reconstruction is quite high.

## 6. Conclusions

In this study, one GWO-SVM model for temperature reconstruction was established based on climate proxies of *Tamarix* cones in the southern edge of the Taklimakan Desert. Using this model, the annual average temperature from 1790 to 2010 AD was reconstructed. This work is beneficial to the proposal and implementation of the proactive measures for environmental sustainability. The conclusions are summarized as follows:

1. NRS is suitable for optimizing climate proxies with a cross redundancy.
2. Utilizing the GWO-SVM model established in this paper, the annual mean temperature in STD can be conveniently and reasonably reconstructed using the climate proxies of *Tamarix* cones.
3. The annual mean temperature in STD has distinct stages during the period from 1790 to 2010 AD, with cold conditions during 1790–1840 AD and 1896–1939 AD, and with warm conditions during 1841–1895 AD and 1940–2010 AD.

**Author Contributions:** Conceptualization, Z.L. and Y.Z.; methodology, Z.L.; software, Z.L. and Z.W. and C.C.; validation, Z.L. and S.Z.; formal analysis, Z.L. and Y.Z.; investigation, Z.L., Z.W. and C.C.; data curation, S.Z.; writing—original draft preparation, Z.L.; writing—review and editing, Z.L. and Y.Z.; supervision, Y.Z.; funding acquisition, Y.Z. All authors have read and agreed to the published version of the manuscript.

**Funding:** This research was funded by the National Natural Science Foundation of China, grant number 41877448.

**Institutional Review Board Statement:** Not applicable.

**Informed Consent Statement:** Not applicable.

**Data Availability Statement:** Research data from this study are available on request (lizhg1978@126.com).

**Conflicts of Interest:** The authors declare no conflict of interest.

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
