# Peer review of "Temperature Reconstruction in the Southern Margin of Taklimakan Desert from Tamarix Cones Using GWO-SVM Model"

_sustainability, doi:10.3390/su151410813_

Round 1

Reviewer 1 Report

Dear authors, I do not need to go into the details of your manuscript because I found it completely out of Climate Change and Environmental Sustainability topic. I strongly recommend to find another topic/theme https://www.mdpi.com/topics in the same journal or select from wide range of MDPI family https://www.mdpi.com/about/journals.

Author Response

Response to Reviewer 1 Comments

Point 1: Dear authors, I do not need to go into the details of your manuscript because I found it completely out of Climate Change and Environmental Sustainability topic. I strongly recommend to find another topic/theme https://www.mdpi.com/topics in the same journal or select from wide range of MDPI family https://www.mdpi.com/about/journals.

Response 1: Thank you for your comments and advice. The purpose of this article is to reconstruct the temperature of the southern edge of Taklamakan for over 200 years. Our temperature reconstruction is consistent with other temperature reconstructions around the southern edge of the Taklimakan Desert. These reconstructed temperatures (including the temperature reconstruction in present work) share the same trend, i.e., the warming from the 19th to the 20th century. Furthermore, this study has identified the characteristics of temperature warming in the southern edge of the Taklamakan region, prompting local people to take more proactive measures to address this situation. In summary, we believe that this article conforms to Climate Change and Environmental Sustainability topic.

Reviewer 2 Report

The study informs about important ecological phynomenas for assessment of climate change impacts. Especially, declared application of CCA is very perspective for multiple assessment of ecosystem response to climate change. However, the study needs improvements of methodical section as well as deep revision of results descripted.

The study needs clear description of methods used. Input separate chapter of Material and methods with characteristics of investigated area, data and statistical analyses. Data description should to be devided between method section and results. Please, characterise a way of CCA and linear trend calculations.

Table 2 and Figure 2 need detail description of abbreviations used except chemical elements. 

Linear trends showed in figure 4 need to be replaced by running means including sufficient definition in methodical section.

Author Response

Response to Reviewer 2 Comments

Point 1: The study needs clear description of methods used. Input separate chapter of Material and methods with characteristics of investigated area, data and statistical analyses. Data description should to be devided between method section and results. Please, characterise a way of CCA and linear trend calculations.

Response 1: Thank you for your comments. We have re-organized the structure of the paper. Chapter 2 of Investigated area and Chapter 3 of Material and methods were added in the revised manuscript. The data source and methods, including Neighborhood rough set model, Support vector machine optimized by grey wolf optimizer, Pearson correlation analyses, and Canonical correlation analyses, were explained in Chapter of Materials and methods. In addition, the linear trend lines were obtained through linear fitting.

Point 2: Table 2 and Figure 2 need detail description of abbreviations used except chemical elements.

Response 2: Figure 2 in the original manuscript has been renamed Figure 6 in the revised manuscript. The detailed descriptions of abbreviations used except chemical elements were added in Table 2 and Figure 6.

Point 3: Linear trends showed in figure 4 need to be replaced by running means including sufficient definition in methodical section.

Response 3: Figure 4 in the original manuscript has been renamed Figure 8 in the revised manuscript. The legend of Figure 8 has been updated. Moreover, the blue solid line in Figure 8 (b) represents the reconstructed temperature anomalies with a 10-year running mean. According to linear fitting, the linear trend (the red dotted line in Figure 8 (b)) of the reconstructed temperature anomalies with a 10-year running mean was obtained.

Reviewer 3 Report

Sustainability_Li

General Comments

The paper examines paleotemperature reconstruction using Tamarix cones. A short-term data set is given and used as the basis for a machine learning approach to extend to a longer time. 

The idea is okay. However, I have one major concern. Not all the 15 measures listed in Table are climate proxies. For example, total C and total N are not climate proxies. Nor do cation concentrations indicate climate. Sorry to be harsh, but you lost me in the description how you used the data sets. The description of the data set, starting on line 87 needs more detail.  

For example, in Table 2, temperature is one signal. But precipitation could be important too. You need to better justify the data set.  

Moreover, correlation is not cause and effect. Thus, on line 151 you state that the chemical parameters can predict mean annual temperature. But logically, mean annual temperature can predict chemistry. Indeed, I would argue that the latter, predicting chemistry from temperature, is more plausible. 

The machine learning is okay to me. The validation against other data sets is helpful.  

Specific Comments

1)    The Abstract is okay. 

2)    The Introduction is okay. However, many readers will not be familiar with the models you used. It would help readers if you justified why you used the models.  

3)    Also, I am not convinced all the measures in Table 1 are climate proxies. For example, carefully read reference 23; it is does not justify the measure as a climate proxy: Nor does reference 27.   

4)    The heart of the paper is the correlation matrix shown in Table 2. I suppose this is okay, but climate is not the only signal. What about precipitation during the period. 

5)    Correlation is a general term for many different statistical procedures. What statistical procedure did you use? Pearson? Spearman Rank?

6)    Figure 2 is okay. However, you need to briefly describe neighborhood rough set theory. The value on the y-axis, Importance, should be clear to readers. It is not.  

Technical Comments

1)    Line 28: delete the first sentence. It is obvious and not necessary.

2)    Line 30: do not use ‘etc.’ in scientific writing. You are making readers guess your thoughts. 

3)    Line 50: the sentence ends abruptly. Either change to ‘nonlinearity’ or finish the thought. 

4)    Line 55: delete the starting clause and start the sentence with ‘the neighborhood rough set, etc.’

5)    Line 82: do you have a better reference than 22, since it is in Chinese, and not available to all readers?

6)    Line 118: reference 10 is in Chinese, so it is not accessible.  

7)    Line 176: it is not clear to me how you reached the conclusion of ‘nonlinear.’ Did you use Spearman Correlation?

8)    Line 237: is one standard deviation arbitrary?  

9)    Line 245: there is no need to list all. Rather give the most interesting deviations.  

The paper is readable.

Author Response

Response to Reviewer 3 Comments

Point 1: The idea is okay. However, I have one major concern. Not all the 15 measures listed in Table are climate proxies. For example, total C and total N are not climate proxies. Nor do cation concentrations indicate climate. Sorry to be harsh, but you lost me in the description how you used the data sets. The description of the data set, starting on line 87 needs more detail.

Response 1: Thank you for your comments. We have carefully reviewed the literature and updated the corresponding references. Many studies have shown that total C, total N, and cation concentrations are closely related to climate [4][20][23]. Thus, total C, total N, and cation concentrations can be considered as climate proxies, to a certain extent.

[4] Xia, X.; Zhao, Y.; Wang, F.; Cao, Q. Environmental significance exploration to Tamarix Cone age layer in Lop Nur Lake region. Chin. Sci. Bull. 200519, 130-131 (in Chinese).

[20] Shen, J.; Wang, Y.; Liu, X. A 16 ka climate record deduced from δ13C and C/N ratio in Qinghai Lake sediments, northeastern Tibetan Plateau. Chin. J. Ocean. Limnol. 2006, 24(2), 103-110.

[23] Lucas, R.W.; Sponseller, R.A.; Laudon, H. Controls Over Base Cation Concentrations in Stream and River Waters: A Long-Term Analysis on the Role of Deposition and Climate. Ecosystems. 2013, 16, 707-721.

Point 2: For example, in Table 2, temperature is one signal. But precipitation could be important too. You need to better justify the data set.

Response 2: Thank you for your comments and advice. The purpose of this article is to reconstruct the temperature of the southern edge of Taklamakan for over 200 years. It is worth noting that precipitation is also one important signal. Therefore, in future work, we will focus on the precipitation reconstruction of the southern edge of Taklamakan based on Tamarix cones.

Point 3: Moreover, correlation is not cause and effect. Thus, on line 151 you state that the chemical parameters can predict mean annual temperature. But logically, mean annual temperature can predict chemistry. Indeed, I would argue that the latter, predicting chemistry from temperature, is more plausible.

Response 3: Thank you for your comments. For a more objective description, we have removed the description, “In general, the annual mean temperature is influenced by many factors”, in the revised manuscript. Undoubtedly, temperature changes will lead to changes in chemical parameters. Based on this fact, we infer the colder and warmer periods in the past years without instrumental data by analyzing the physical and chemical indicators of Tamarix cones.

Point 4: The Introduction is okay. However, many readers will not be familiar with the models you used. It would help readers if you justified why you used the models.

Response 4: The introduction of the models used in this paper was added in Section 3.2 in the revised manuscript. Moreover, the reasons for using these models were described in Section Introduction.

Point 5: Also, I am not convinced all the measures in Table 1 are climate proxies. For example, carefully read reference 23; it is does not justify the measure as a climate proxy: Nor does reference 27.

Response 5: Thank you for your comments. We have carefully reviewed the literature and updated the corresponding references. Many studies have shown that total C, total N, and cation concentrations are closely related to climate [4][20][23]. Thus, total C, total N, and cation concentrations can be considered as climate proxies, to a certain extent.

[4] Xia, X.; Zhao, Y.; Wang, F.; Cao, Q. Environmental significance exploration to Tamarix Cone age layer in Lop Nur Lake region. Chin. Sci. Bull. 200519, 130-131 (in Chinese).

[20] Shen, J.; Wang, Y.; Liu, X. A 16 ka climate record deduced from δ13C and C/N ratio in Qinghai Lake sediments, northeastern Tibetan Plateau. Chin. J. Ocean. Limnol. 2006, 24(2), 103-110.

[23] Lucas, R.W.; Sponseller, R.A.; Laudon, H. Controls Over Base Cation Concentrations in Stream and River Waters: A Long-Term Analysis on the Role of Deposition and Climate. Ecosystems. 2013, 16, 707-721.

Point 6: The heart of the paper is the correlation matrix shown in Table 2. I suppose this is okay, but climate is not the only signal. What about precipitation during the period.

Response 6: Thank you for your comments and advice. The purpose of this article is to reconstruct the temperature of the southern edge of Taklamakan for over 200 years. It is worth noting that precipitation is also one important signal. Therefore, in future work, we will focus on the precipitation reconstruction of the southern edge of Taklamakan based on Tamarix cones.

Point 7: Correlation is a general term for many different statistical procedures. What statistical procedure did you use? Pearson? Spearman Rank?

Response 7: The introduction of statistical analyses was added in Section 3.2 in the revised manuscript. In this paper, the correlations between climate proxies and temperature were determined using Pearson correlation analyses and Canonical correlation analyses from IBM SPSS Statistics 2.4, respectively.

Point 8: Figure 2 is okay. However, you need to briefly describe neighborhood rough set theory. The value on the y-axis, Importance, should be clear to readers. It is not.

Response 8: Figure 2 in the original manuscript has been renamed Figure 6 in the revised manuscript. The introduction of neighborhood rough set theory was added in Section 3.2 in the revised manuscript, which can make the value on the y-axis, Importance, clear to readers.

Point 9: Line 28: delete the first sentence. It is obvious and not necessary.

Response 9: The first sentence in Section introduction has been deleted.

Point 10: Line 30: do not use ‘etc.’ in scientific writing. You are making readers guess your thoughts.

Response 10: We have deleted ‘etc.’ in the corresponding sentence.

Point 11: Line 50: the sentence ends abruptly. Either change to ‘nonlinearity’ or finish the thought.

Response 11: We have deleted the corresponding sentence.

Point 12: Line 55: delete the starting clause and start the sentence with ‘the neighborhood rough set, etc.’

Response 12: We have deleted the corresponding sentence.

Point 13: Line 82: do you have a better reference than 22, since it is in Chinese, and not available to all readers?

Response 13: Reference 22 has not been published in English journals. In order to help readers understand the chronological sequence of the Tamarix cones, Figure 2 which was re-organized from reference 22 was added in the revised manuscript.

Point 14: Line 118: reference 10 is in Chinese, so it is not accessible.

Response 14: Reference 10 has not been published in English journals, but the owner of the data is the corresponding author of this article. If you need any additional information on the paper, please feel free to contact me (lizhg1978@126.com).

Point 15: Line 176: it is not clear to me how you reached the conclusion of ‘nonlinear.’ Did you use Spearman Correlation?

Response 15: The conclusion of ‘nonlinear’ is summarized from relevant literature about the relationship between climate proxies and temperature. The corresponding references are as follows.

[33] Emile-Geay, J.; Tingley, M. Inferring climate variability from nonlinear proxies: application to palaeo-ENSO studies. Clim. Past. 201511(4), 31-50.

[34] Bauwens, M.; Ohlsson, H.; K Barbé; Beelaerts, V.; Dehairs, F.; Schoukens, J. On climate reconstruction using bivalves: Three methods to interpret the chemical signature of a shell. Computer Methods and Programs in Biomedicine. 2011, 104(2), 104-111.

[35] Zhang, Z.; Wagner, S.; Klockmann, M.; Zorita, E. Evaluation of statistical climate reconstruction methods based on pseudoproxy experiments using linear and machine-learning methods. Clim. Past. 2022, 18, 2643-2668.

[36] Kaufmann, G.; Dreybrodt, W. Stalagmite growth and palaeo-climate: an inverse approach. Earth and Planetary Science Letters. 2004, 224(3-4), 529-545.

Point 16: Line 237: is one standard deviation arbitrary?

Response 16: The standard deviation of the reconstructed temperature was added in Section 5.1 in the revised manuscript.

Point 17: Line 245: there is no need to list all. Rather give the most interesting deviations.

Response 17: The related description has been re-written in an attempt to provide more interesting points.

Round 2

Reviewer 1 Report

Dear authors, on one hand, I am very glad of the improvements of your text and more explaining the methodology. On other hand, the responds did not convince me that your manuscript could be part of special volume “Climate Change and Environmental Sustainability, 2nd Volume”. Quotation: “The Topic “Climate Change and Environmental Sustainability II” welcomes high-quality works focusing on the development and implementation of systems, ideas, pathways, solutions, strategies, technologies, and pilot cases and exemplars that are relevant to climate change impact measurement and assessment, mitigation and adaptation strategies and techniques, public participation and governance.”

I do not found any of these feature in your manuscript, please, try to find another MDPI journal or specific issues that corresponds to your findings.

Author Response

Point 1: The responds did not convince me that your manuscript could be part of special volume “Climate Change and Environmental Sustainability, 2nd Volume”. Quotation: “The Topic “Climate Change and Environmental Sustainability II” welcomes high-quality works focusing on the development and implementation of systems, ideas, pathways, solutions, strategies, technologies, and pilot cases and exemplars that are relevant to climate change impact measurement and assessment, mitigation and adaptation strategies and techniques, public participation and governance”. I do not found any of these feature in your manuscript, please, try to find another MDPI journal or specific issues that corresponds to your findings.

Response 1: Thank you for your comments and advice. The purpose of this manuscript is to reconstruct the temperature of the southern edge of Taklamakan over the past 200 years. Recently, Jessica Tierney of the University of Arizona and her collaborators published a review paper in the journal Science, systematically summarizing and evaluating the important role of paleoclimate information in scientific response to future climate change, and emphasizing its importance for modern climate simulation and future climate change prediction [1]. Therefore, this manuscript belongs to “Climate change prediction and analysis” in the Theme “Climate Change Impact Assessment” of the Topic “Climate Change and Environmental Sustainability II”. To illustrate the contribution of our work to climate prediction and environmental sustainability, we have added corresponding descriptions in the Abstract, Introduction, and Conclusion Sections. The details are as follows:

Line 21-24 in Abstract Sections: “The present work is beneficial to predicting the future climate in the local area and encouraging local governments to develop more effective measures to address the risks of climate change to environmental sustainability.”

Line 33-38 in Introduction Sections: “Due to the close relationship between ecological environment and climate, the climate change can be used to effectively characterize the ecological environment. Indeed, paleoclimate reconstruction plays a vital role in climate prediction and then contributes to the proposal and implementation of more proactive measures to address potential environmental sustainability risks brought about by climate change, such as education, publicity, and policy adjustments to reducing carbon emissions [1].”

Line 414-419 in Conclusion Sections: “In this study, one GWO-SVM model for temperature reconstruction was established based on climate proxies of Tamarix cones in southern edge of the Taklimakan Desert. Using this model, the annual average temperature from 1790 to 2010 AD was reconstructed. This work is beneficial to the proposal and implementation of the proactive measures for environmental sustainability. The conclusions are summarized as follows:”

[1]Tierney J, Poulsen C, Montanez I et al. Past climates inform our future [J]. Science, 2020, 30(6517).

Reviewer 2 Report

The authors accepted all the reviewer recommendations. Especially, the authors input separated chapters about investigated area including map and detail description of methods with quantity definitions around all formulas. Additionally, the result and discussion sections were improved in relation to fitting of the time-series presented. Thus now, the article seems as ready for publication. 

Author Response

Point 1: The authors accepted all the reviewer recommendations. Especially, the authors input separated chapters about investigated area including map and detail description of methods with quantity definitions around all formulas. Additionally, the result and discussion sections were improved in relation to fitting of the time-series presented. Thus now, the article seems as ready for publication.

Response 1: Thank you for your comments and advice. We gratefully appreciate the comments, which are very useful input about our paper.

Round 3

Reviewer 1 Report

Dear authors, I do agree that paleoclimatology researches opens new perspectives fro climate change studies and creates background for future predictions. I still think, that your research could reach broader audience and make higher impact while being publish in different journal or book series. However, some additional text makes it more feasible for the topicality of the special issue.